# Molecular Pathogenesis and Immune Evasion of Vesicular Stomatitis New Jersey Virus Inferred from Genes Expression Changes in Infected Porcine Macrophages

**DOI:** 10.3390/pathogens10091134

**Published:** 2021-09-03

**Authors:** Lauro Velazquez-Salinas, Jessica A. Canter, James J. Zhu, Luis L. Rodriguez

**Affiliations:** 1Foreign Animal Disease Research Unit, Plum Island Animal Disease Center, United States Department of Agriculture–Agricultural Research Service, Greenport, NY 11944, USA; Jessica.Canter@usda.gov; 2Plum Island Animal Disease Center Research Participation Program, Oak Ridge Institute for Science and Education, Oak Ridge, TN 37830, USA

**Keywords:** vesicular stomatitis virus, macrophage, microarray analysis, differential gene expression, molecular pathogenesis, immune evasion

## Abstract

The molecular mechanisms associated with the pathogenesis of vesicular stomatitis virus (VSV) in livestock remain poorly understood. Several studies have highlighted the relevant role of macrophages in controlling the systemic dissemination of VSV during infection in different animal models, including mice, cattle, and pigs. To gain more insight into the molecular mechanisms used by VSV to impair the immune response in macrophages, we used microarrays to determine the transcriptomic changes produced by VSV infection in primary cultures of porcine macrophages. The results indicated that VSV infection induced the massive expression of multiple anorexic, pyrogenic, proinflammatory, and immunosuppressive genes. Overall, the interferon (IFN) response appeared to be suppressed, leading to the absence of stimulation of interferon-stimulated genes (ISG). Interestingly, VSV infection promoted the expression of several genes known to downregulate the expression of IFNβ. This represents an alternate mechanism for VSV control of the IFN response, beyond the recognized mechanisms mediated by the matrix protein. Although there was no significant differential gene expression in macrophages infected with a highly virulent epidemic strain compared to a less virulent endemic strain, the endemic strain consistently induced higher expression of all upregulated cytokines and chemokines. Collectively, this study provides novel insights into VSV molecular pathogenesis and immune evasion that warrant further investigation.

## 1. Introduction 

Vesicular stomatitis virus (VSV) infection causes fever and vesicular stomatitis, one of four clinically indistinguishable viral vesicular diseases. VSV (family Rhabdoviridae, genus Vesiculovirus) is comprised of a non-segmented RNA viral genome encoding five structural proteins: nucleocapsid (N), phosphoprotein (P), matrix (M), glycoprotein (G), and the large RNA-dependent RNA polymerase (L) [1,2], along with two non-structural proteins (C and C0) of undetermined function encoded in overlapping reading frames of the P gene [3]. VSV causes most of the cases of vesicular diseases reported in livestock. resulting in economic losses associated with quarantines imposed by animal health authorities due to its similar clinical presentation with foot and mouth disease virus (FMDV) [4,5]. 

VSV has a broad host range and cell tropism due to its glycoprotein binding to host LDLR family members that are ubiquitously expressed on host cells and conserved among mammalian species [6,7]. Along with typical vesicular lesions in specific tissues, infected animals also show systemic signs such as anorexia, lethargy, and fever (https://en.wikivet.net/Vesicular_Stomatitis_Virus (accessed on 26 July 2021)). Despite these clinical signs, VSV infection typically does not result in host mortality [4,5]. After infection via insect bites, animals show limited virus replication, primarily in specific tissues where the vesicular lesions occur. Infected animals usually recover completely within 2–3 weeks of infection [4,5]. 

Unlike gross pathogenesis, the molecular pathogenesis of VSV is not very clear. On the basis of a literature review, it appears that only TNF has been investigated in VSV pathogenesis, showing more rapid induction of TNF by an attenuated VSV mutant after infection but more drastic TNF induction later in infection by wild-type VSV in mice [8]. TNF knockout mice showed diminished weight loss following wild-type VSV infection, and the rapid weight loss seen in wild-type VSV infection was less pronounced in C57BL/6 mice infected by an attenuated mutant virus [8]. In mice, interferons produced by VSV-infected macrophages play a key role in protection against the neuropathogenesis of the virus [9]. In the natural hosts such as cattle, VSV antigens were colocalized with an antibody against a marker molecule (MAC387, MRP14, or S100A9) of myeloid cells, including macrophages, using immunohistochemistry [10]. Both wild-type VSV and matrix protein mutants productively replicate in porcine immune cells and non-immune cells [11,12]. Infection with wild-type VSV induced weaker proinflammatory cytokine responses and downregulated the expression of the costimulatory molecule complex CD80/86 and MHC class II compared to the matrix protein mutant virus [11]. A matrix protein (M51R) VSV mutant virus replicated ~1000 times less in cultured primary porcine macrophages than its wild-type counterpart and showed significantly diminished virulence in pigs [13]. The molecular pathogenesis and immune evasion in natural hosts such as pigs and cattle have yet to be investigated. 

It is well-known that VSV can inhibit the host interferon response primarily via its matrix proteins [14,15]. VSV matrix protein mRNA can be translated into three proteins starting at three in-frame start codons [16]. Transfection with plasmids containing the M protein gene alone can induce CPE in transfected cells [16]. VSV M proteins can delay apoptosis induced by other viral components [17] and suppress transcription in infected cells by inhibiting the basal transcription factors TFIID and TFIIH and interacting with host Rae1 and Nup98 [18,19,20,21,22]. VSV M proteins can also inhibit the nuclear export of host mRNA and snRNAs [23] and NFκB activation [24]. The suppression of IFNβ expression by the matrix protein is correlated with the inhibition of host RNA and protein synthesis [25]. A systems biology approach including transcriptomic analysis has been conducted to study VSV infection in a murine macrophage cell line [26,27]; however, VSV pathogenesis and immune evasion were not explicitly explored on the basis of transcriptional changes after virus infection. Although mice have been extensively used as an experimental model for VSV infection, they are not natural hosts for VSV infection. The transcriptomic analysis of VSV infection has not been investigated in the primary macrophages of its natural livestock hosts.

Macrophages play an important role in host defense against pathogens via positioning in all tissues, where they can effectively sense danger signals with highly expressed PAMP receptors and produce a large quantity of both pro- and anti-inflammatory cytokines, such as IL-1, IL-10, TGFβ, and TNF, via cell polarization and differentiation to regulate the immune response [28,29]. Our previous study showed that primary porcine macrophages expressed higher levels of IFNβ and cytokines than primary fetal porcine kidney cell cultures after VSV infection [13]. Given that VSV can infect and replicate in macrophages and the important role of macrophages in the immune response, ex vivo porcine macrophages were used as model cells to extrapolate the molecular mechanisms of VSV pathogenesis and immune evasion. The objective of this study was to formulate hypotheses for the molecular mechanisms of VSV pathogenesis and immune evasion on the basis of gene expression changes in porcine macrophages after infection for further investigation. 

## 2. Results 

### 2.1. Differential Gene Expression

There were no genes differentially expressed between macrophages infected with epidemic VS New Jersey (NJ0612NME6) and endemic New Jersey (NJ0806VCB) strains (minimal FDR = 0.13). There was a total of 4346 significant differentially expressed genes at a false discovery rate (FDR) of ≤0.05 with at least a 50% difference and a total of 3345 with a difference of 2-fold or greater between epidemic VSV and mock-infected macrophages. Between epidemic VSV and mock-infected macrophages, there was a total of 3345 significant differentially expressed genes (DEGs) by at least 50% at a false discovery rate (FDR) of ≤0.05. Among these DEGs, the majority were detected as being downregulated (2179 DEGs) compared to 1166 upregulated genes between VSV-infected and mock-infected cells, which was at approximately a 2:1 ratio. Forty-four percent of DEGs were differentially expressed by 1.5- to 2.0-fold, with 841 of these genes being downregulated and 618 being upregulated (Figure 1). There were 54% of DEGs with a fold change between 2.0 and 5.0, and the largest proportion of genes, 1288, were downregulated compared to 521 upregulated genes. Finally, the remaining 2.3% of the DEGs were differentially expressed with a fold change greater than 5 (27 genes downregulated and 50 genes upregulated) (Figure 1). The most drastic differences were at a fold change of 10.2 for a downregulated gene and 32.8 for an upregulated gene between these VSV- and mock-infected macrophages. 

### 2.2. Pathway Analyses

To identify the biological pathways/processes most impacted by the differential expression, the lists of DEGs with differential expression of a fold change of at least 2-fold to remove DEGs with minor effects were used in the DAVID analysis. GO term analysis showed that NFκB signaling pathway was the most over-represented by the DEGs, with three other significant biological processes in protein ubiquitination, Toll-like receptor signaling, and mRNA transcription regulation (Table 1). KEGG pathway analysis identified eleven over-represented biological pathways with five top pathways (TNF, TLR, NFkB, RIG-I-like receptor, and NOD-like receptor signaling) that are known to play key roles in the immune response (Table 1). Only one biological pathway (TNF-induced apoptosis) was detected with REACTOME analysis.

When this list of DEGs was further narrowed to those with a fold change of at least 4 (more biologically impactful DEGs), GO term analysis identified thirteen over-represented biological processes; six involved in the immune response, four in apoptosis, two in signaling pathways, and one in RNA transcription. The two most over-represented GO terms were in inflammatory response (GO_0006954) and apoptotic process (GO_0006915) (Table 1). The significantly over-represented KEGG pathways included the KEGG_hsa04060-cytokine–cytokine receptor interaction, the KEGG_hsa04668-TNF-signaling pathway, the KEGG_hsa04064-NF-kappa B-signaling pathway. the KEGG_hsa04621-NOD-like receptor signaling pathway, and KEGG_hsa04620-Toll-like receptor signaling pathway. Only one Reactome pathway (HSA-380108: Chemokine receptors bind chemokines) was significant. 

### 2.3. Interferon Expression and Signaling

VSV infection significantly induced the expression of IFNB by 6.2-fold but did not induce expression of other interferons in infected macrophages compared to mock-infected cells (Table 2). The endemic strain induced higher IFNB expression (2-fold) than the epidemic strain but not at a significant level. VSV infections significantly suppressed the expression of an IFNA homologous to human IFNA17 by approximately 2-fold and did not significantly alter the expression of other interferons, including types II and III. The expression of type I (IFNAR1 and IFNAR2) and II (IFNGR1) receptors were suppressed in VSV-infected cells compared to mock-infection (Table 2). The type III IFN receptor (IFNLR1) was expressed at a very low level in the macrophages (signal intensity = 69, SNR < 2). The expression of typical interferon-stimulated genes (ISGs) was not significantly changed by VSV-infection (only 10 genes listed in Table 2). These results indicate that VSV infection suppressed type I IFN and II IFN signaling. 

Six genes, AHR [30,31], ATF3 [32], DUSP1 [33], FOS [34], HES1 [35], and PRDM1 [36], known to negatively regulate type I IFN expression, were significantly induced mostly by >10-fold in VSV-infected cells (Table 2). Among all DEGs, PRDM1 was the most induced gene (~33-fold higher) after infection, and FOS was the fifth-most-induced gene in this study (Table 7). EGR1, a PRDM1 expression-inducing gene [37], was also highly upregulated (13.4-fold) in VSV-infected cells. These results indicate that VSV infection induces the expression of genes suppressing IFNB expression.

### 2.4. Immune Signaling Pathways

The expression of a transcription factor (ATF2) and five MAPK kinases (MAP2K5, MAPK14/p38, MAPK4, and MAP3K18) in the MAPK signaling pathways was significantly downregulated in VSV-infected cells compared to mock-infected cells (Table 3). Likewise, the expression of seven activator genes (CARD6, IKBKB, IRAK1, NLK, TAB1, TAB2, and TAK1) in the NFκB pathway was significantly downregulated in VSV-infected cells, whereas the expression of three inhibitors of NFκB (NFKBIA, NFKBID, and TNFIP3/A20 [38]) was significantly upregulated (Table 3). Three genes (IRF5, MAVS, and TBK1) in the RIG-I signaling pathway were expressed at significantly higher levels in VSV-infected cells than in mock-infected cells (Table 3). The expression of four TLR receptors (TLR1, TLR2, TLR4, and TLR6) and two signal transducers (BTK and TRIF) was downregulated in VSV-infected cells compared to mock-infected cells, whereas TLR7 was upregulated (Table 3). These results indicate that VSV infection suppresses the signaling of the MAPK, NFκB, RIG-I, and TLR pathways.

### 2.5. Cytokines, Chemokines, and Receptors

VSV infection significantly induced the expression of six immune cytokines (CSF3, IL1A, IL10, IL27, TNF, and TNFSF9) and suppressed TNFSF11 expression (Table 4). Four non-typical immune cytokines (AREG, HBEGF, LIF, and VEGFA) were expressed at significantly higher levels in VSV-infected than mock-infected cells (Table 4). Among those cytokines, AREG, IL1A, IL10, LIF, and TNF were upregulated by >11-fold. Overall, the endemic strain induced consistently higher expression (averaging 1.7-fold) of the upregulated cytokines than the epidemic strain, though at not significant levels, whereas the receptor expression was very similar (Table 4). There were three significantly downregulated (IL17RA, LTBR, and TNFRSF1A) and three upregulated (TNFRSF10, IL1R2, and IL20RB) cytokine receptors in VSV-infected cells compared to mock-infected cells (Table 4). All these DEGs are proinflammatory genes with the exception of IL10, IL1R2, IL20RB, and the four non-typical immune cytokines. These results show that VSV infection induced both pro- and anti-inflammatory cytokine expression and suppressed the expression of IL-17 and TNF receptors.

VSV infection significantly induced the expression of seven chemokines (CCL3, CCL4, CCL5, CCL20, CXCL1, CXCL2, and CXCL3) by ~3- to 42-fold compared to mock infection (Table 5). As for cytokines, the endemic strain also induced higher expression of the upregulated chemokines (by 1.6-fold) than the epidemic strain, though not significantly, and nearly identical receptor expression (Table 5). The expression of three chemokine receptors (CCR5 (the receptor of CCL3, CCL4, and CCL5), CCLRL2, and CX3CR1) was significantly downregulated by VSV infection (Table 5), whereas CCR7 and CXCR4 expression was significantly induced after VSV infection when compared to mock-infected cells (Table 5). The results suggest that chemokines upregulated by VSV infection could cause infiltration of neutrophils, macrophages, and Th17 cells in the infected tissue according to their chemotactic activities [39]. On other hand, the infection could also alter the response of the infected cells to chemokines.

### 2.6. Apoptosis, Autophagy, and Unfold Protein Response

The expression of three pro-apoptotic genes, BCL2L13 [40], DAPK1 [41], and DIDO1 [42], and a key caspase (CASP8) in the apoptosis-activating pathway were significantly downregulated in VSV-infected macrophages (Table 6). On the other hand, two apoptosis inhibitors, BIRC3/cIAP2 and SGK1 [43], and an activator of the apoptosis inhibitor expression, REL [44], were upregulated (Table 6). Two negative regulator genes of TNF-induced apoptosis, BRE [45] and IER3 [46], were also downregulated in VSV-infected cells (Table 6). 

The expression of eight autophagy-associated genes, including seven ATGs and FLCN [47], and three positive autophagy regulators RB1CC1 [48], Rab33b [49] and ULK1 [47] was significantly lower in VSV-infected cells than in mock-infected cells (Table 6). Two autophagy inhibitors, BCL2L11/BIM [50] and Gadd45b [51], were expressed at significantly higher levels in VSV-infected cells than in mock-infected cells (Table 6). GADD45B was one of the top 10 most-induced genes after VSV infection with >20 fold upregulation. 

The expression of XBP1, a key regulator in stress-induced unfolded protein response (UPR) [52], and ERN1, the ER stress sensor of UPR [53], was significantly downregulated in VSV-infected cells compared to mock-infected cells (Table 6). PPP1R15A (GADD34) mediates dephosphorylation of eIF2alpha in a negative feedback loop and inhibits the unfolded protein response (UPR) [54], and its expression was significantly upregulated by 7-fold in VSV-infected macrophages compared to mock infection (Table 6). The results of gene expression changes after infection suggested that VSV suppresses apoptosis, autophagy, and the UPR response. 

### 2.7. Host mRNA Transcription, Modification, and Stability

Thirteen genes involved in the transcription of host RNA based on KEGG pathways were significantly downregulated in VSV-infected cells compared to mock-infected macrophages (Table 7). The expression of five genes (CMTR2 (mRNA cap methylation), DICER1 (microRNA processing), MIR132 (microRNA), TIAL1 (selective binding to several mRNAs to control expression of translation regulatory proteins) [55], and ZFP36/TTP (AT-rich mRNA degradation) [56]) in post-transcriptional mRNA processing were significantly affected by VSV infection and were the first two downregulated and the last two upregulated, respectively (Table 7). These results suggested that host mRNA transcription and processing could be negatively impacted by VSV infection to facilitate VSV replication.

### 2.8. Inflammation-Related Genes

Two proinflammatory mediator genes, ADM [57] and a prostaglandin E (PGE) synthase (PTGS2) [58] were expressed at significantly higher levels (5.6 and 15.6 times, respectively) after VSV infection (Table 8). MALAT1 and DUSP2 play a role in prostaglandin E2 production in macrophages [59,60], and their expression was also upregulated by VSV infection (Table 8). There were three C5a or purinergic receptor genes involved in macrophage M1 activation, including C5AR1 [61], P2RY1, and P2RY6 [62] being downregulated and a suppressor (IL4I1) of macrophage M1 activation [63] upregulated in VSV-infected cells (Table 8).

A multitude of genes associated with pro-inflammatory responses was also differentially affected by VSV infection compared to mutant vs. mock infection (Table 8). Two genes, MAP3K8 [64] and MEFV [65], critical for IL-1 and TNF production, were upregulated in VSV-infected cells. On the other hand, four proinflammatory genes (CEBPD, DUSP6, MAFB, and MAPK8IP3/JIP3) associated with the regulation of the NFkb pathway [66,67,68,69] and one gene (MTRES1) implicated in the stress response in mitochondria [70] were downregulated during the VSV infection. These results indicate VSV can alter host gene expression via both up- and downregulation of proinflammatory genes. 

## 3. Discussion 

The molecular mechanisms of VSV pathogenesis remain unclear. Multiple studies suggest that interaction with immune cells and regulation of the immune response play a role in determining the outcome of infection [10,11,12,13,14,15]. Because of the important roles of immune cytokines in the pathogenesis of viral infection [29] and macrophages in the production of proinflammatory cytokines and their wide tissue distribution [28], VSV-infected porcine macrophages are an excellent model to extrapolate the molecular mechanisms of VSV pathogenesis. Our transcriptomic analysis shows that VSV infection induces massive (>10-fold) expression of proinflammatory cytokines, including IL1A and TNF, especially TNF (23-fold); chemokines, e.g., CCL4, CCL5, and CXCL2 (>20 times); and a PGE synthetase gene (PGTS2). CCL3, CCL4, and CCL5 share the same receptor (CCR5). Likewise, CXCL1 and CXCL3 also share the same receptor (CXCR1) with CXCL2 [39]. These chemokines were also massively induced after VSV infection. High production of these cytokines and chemokines and PGE is known to induce fever [71,72,73,74]. Interestingly, LIF expression was highly upregulated after VSV infection, and LIF injection can induce fever in animals [75,76]. On the basis of these results, it is hypothesized that high fever is mainly caused by VSV-induced high production of CCR5 and CXCR1 ligands, IL1A, LIF, PGE, and TNF. These cytokines, chemokines, and PGE are potent mediators of inflammation. IL1A and TNF are well-known potent proinflammatory cytokines. PGE induces vasodilatation and local recruitment of neutrophils, macrophages, and mast cells at early stages of inflammation [77,78]. Chemokines induced by VSV infection, including CCL3, CCL4, CCL5, CCL20, CXCL1, CXCL2, and CXCL3, recruit macrophages, NK cells, neutrophils, and/or Th17 cells [39]. CSF3 stimulates neutrophil generation in the bone marrow [79]. The expression level of CSF3 was significantly increased after VSV infection. Neutrophils are known to play a key role in the clearance of viruses via phagocytosis and neutrophil extracellular traps [80]. Therefore, the high production of PGE, cytokines, and chemokines by VSV-infected cells could play a role in the pathogenesis within infected tissues by recruiting proinflammatory immune cells and inhibiting viral infection and spread.

TNF is one of the most potent proinflammatory cytokines, and it also induces cell death via apoptosis, necroptosis, and pyroptosis pathways [81]. Two genes, MAP3K8 [64] and MEFV [65], critical for TNF production, were upregulated after VSV infection. Additionally, TIAL1 binds to the 3′ end noncoding sequences of several mRNAs, such as eIF4A, eIF4E, eEF1B, and c-Myc, to control the expression of translation regulatory proteins, which repress protein biosynthesis in cells responding to stress [55]. TIAL1-deficient mice develop arthritis and elevated TNF expression [82]. Upregulated MAP3K8 and MEFV and downregulated TIAL1 in VSV-infected cells strongly support a very important role of TNF both in VSV local and systemic pathogenesis.

Other cytokines highly induced by >10-fold include AREG, IL-10, and LIF, which are known to have immune-suppressive effects. High levels of IL-10 suppress the innate and adaptive immune responses [83]. TNF leads to IL-10 production by monocytes and, together with IL-10, inhibits CD4 T-cell expansion and function [84]. LIF can suppress IFNγ and LPS signaling [85,86]. LIF appears to be an immune-tolerogenic cytokine based on promoting Treg differentiation and inhibiting pro-inflammatory Th17 cell differentiation [87]. Several growth factors, such as VEGF and EGFs, can inhibit IFNB expression [35] or suppress the anti-VSV activity of IFNα and IFNβ [88]. The expression of VEGF and two EGFs (AREG and HBEGF) were induced after VSV infection in this study. PGE_2_ selectively suppresses effector functions of macrophages and neutrophils and Th1-, CTL-, and NK cell-mediated type 1 immunity, but it promotes Th2, Th17, and regulatory T cell responses [77,78]. Therefore, we hypothesize that high levels of EGFs, IL-10, LIF, PGE, and VEGF could play a key role in suppressing the immune response of infected and non-infected cells to facilitate VSV infection and cause disease. 

Our results also indicate that VSV can evade the immune response of infected cells by various mechanisms. It is known that VSV can activate IFN response via RIG-I-MAVS and TLR4/CD14 signaling pathways to induce an antiviral response [14,89]. The expression of two key signaling transducers, TBK1 and MAVS [90,91], in the RIG-I-MAVS signaling pathway was downregulated in VSV-infected cells compared to mock-infected cells. It has been previously reported that VSV glycoprotein binds to the TLR4/CD14 dimer, leading to the induction of interferon expression, mainly mediated by IFNB via a TICAM1/TICAM2-dependent but MyD88- and NFκB-independent signaling pathway [92]. In our study, the expression of TLR4 and TICAM2 was significantly suppressed by VSV infection. Infection of monocytes by VSV has been reported to suppress type I IFN and cytokine (IL-27 and TNF) responses in a viral RNA-specific and TLR7-dependent pathway [93], and TLR7−/− mice show significantly reduced VSV titers in the draining lymph nodes and diminished viral replication in subcapsular sinus macrophages [94]. Our results showed that TLR4 and TLR7 expression was significantly upregulated in both VSV- and mutant-infected cells (Table 7). Therefore, VSV-altered expression of genes in the virus-sensing pathways could be another immune-evading mechanism in infected cells.

Previous studies of IFNB promoters showed that ATF2-JUN, IRF, and NFκB transcription factors regulate IFNB expression [95]. The expression of ATF2 and IRF5 transcription factors was downregulated by VSV-infection. There were several downregulated signaling transducer or upregulated signaling inhibitor genes that could inhibit MAPK and NFkB signaling pathways (Table 2). Interestingly, there were six suppressor genes of IFNB expression (AHR, ATF3, DUSP1, FOS, HES1, and PRDM1) upregulated by up to 32.8-fold in VSV-infected cells. This result supports published results that VSV suppresses the interferon response.

Our results show that VSV infection did not induce expression of other type I interferons except IFNB in pig macrophages. IFNB induction is known to induce IRF7 expression, which is needed for the induction of IFNA [96]. Although IFNB expression was induced in infected macrophages, the expression of IRF7 and IFNA was not increased by VSV infection. This could be explained by the suppression of interferon signaling mediated by downregulated expression of type I and II interferon receptors, as shown in Table 6. This seems to be a novel immune evasion mechanism of VSV in addition to the inhibition of mRNA nuclear export mediated by VSV matrix protein [14,15]. It has been previously reported that VSV infection inhibits the expression of interferon-stimulated genes via miR-132 to facilitate viral replication [97,98]. We found that miR-132 is highly upregulated by VSV infection (Table 7), which could also explain the lack of induction of interferon-stimulated genes by IFNB.

Viral infections trigger three inter-connected biological processes, including apoptosis, autophagy, and stress-induced unfolded protein response (UPR), which can inhibit virus replication [52]. However, viruses can subvert or even manipulate these responses to promote infection; for example, VSV can delay the onset of apoptosis [17,99]. Our results show that multiple genes associated with these three processes were differentially expressed during VSV infection, as listed in Table 6. Among these genes, GADD45B, which is one of the top 10 most-induced genes (22.3-fold) after VSV infection, suppresses apoptosis and autophagy [51]. The expression of two key regulatory genes, XBP and PPP1R15A, in the UPR pathway was reduced after VSV infection. Additionally, genes in death receptor signaling, including TNFRSF1A and several signaling transducers, were downregulated in VSV-infected cells, which could delay the necroptosis/apoptosis induced by high expression of TNF. These results suggest that VSV can suppress these three important innate immune mechanisms during infection.

The chemokines associated with the CCR5 receptor activate macrophages to induce pro-inflammatory cytokine expression [100,101]. Expression of CCR5 was significantly downregulated in VSV-infected macrophages, potentially reducing the immunostimulatory effect of CCR5. Additionally, high IL10 and LIF expression and downregulation of IL17RA in the infected cells could mitigate the effects of Th17 cells recruited by increased CCL20 expression. Therefore, VSV appears to be able to evade the immune response associated with chemokines induced by its infection in infected cells. Published results showed that M2 macrophages were more susceptible to infection and killing by both wild-type and an M51R-M VSV mutant than M1 macrophages [102]. The M1 and M2 activation of macrophages play important roles in the innate and adaptive immune responses. Our results suggest that VSV infection could suppress M1 activation by downregulating the expression of several purinergic receptors, such as C5AR1, CCR5, IL17RA, P2RY1, and P2RY6, and by upregulating IL4I1 expression.

It has been reported that VSV interferes with host gene transcription [19]. Our results showed that VSV downregulated several genes of the host transcription machinery, as shown in Table 7. Additionally, VSV may control host protein translation by altering the expression of genes involved in maintaining mRNA stability and cap-modification, microRNA processing, and protein translation. VSV encodes proteins with cap methylation activity, and mutants lacking this activity show attenuated virulence [103]. Our results showed downregulation of CMTR2, which could negatively impact host translation. ZFP36/TTP is known for its central role in destabilizing mRNA molecules containing class II AU-rich elements in 3’ untranslated regions [56], which are frequently found in cytokine mRNA 3′-end non-coding sequences. Increased expression of this gene could destabilize cytokine mRNA and reduce its translation. Dicer1 was downregulated by VSV infection in this study, and previous works have demonstrated that Dicer1-deficient mice are hyper-susceptible to VSV infection [104]. VSV infection induced a 7-fold increase of PPP1R15A (GADD34) in our study; this protein mediates dephosphorylation of eIF2α, which inhibits viral replication [105,106] and suppresses UPR [54].

In summary, VSNJV infection significantly induced the massive expression (>10-fold) of pro-inflammatory cytokines (IL1A and TNF), chemokines (CCL4, CCL20, and CXCL2) and prostaglandin E and upregulated PTGS2 and immune-suppressive cytokines (IL10 and LIF), which are known to induce fever, immune suppression, and/or recruitment of immune cells. It is hypothesized that these cytokines, chemokines, and possibly PGE play important roles in local and systemic VSV pathogenesis and immune evasion. Although non-significant DEGs were seen between epidemic and endemic VSV strains, the endemic strain consistently induced higher expression of all upregulated cytokines and chemokines (Table 4). These differences are consistent with the overall amino acid differences at N (n = 1), P (n = 6), G (n = 3), and L (n = 9) between NJ0612NME6 and NJ0806VCB strains and might help explain the differences in virulence previously observed in pigs [12]. At this point, on the basis of a previous study indicating the relevance of glycoprotein in the virulence of VSV [107], we may speculate that changes at this level may affect the cytokine profile between both strains. Currently, studies using reverse genetics are being performed in our laboratory to assess the potential role of specific amino acid variations in differences in the virulence between both strains. 

Based on our results, the mechanisms of VSV immune evasion could be achieved via suppressing (a) IFNβ expression; (b) type I and II interferon, IL-1, and death receptor signaling and the TLR and RIG-I signaling pathways; (c) biological processes involved in apoptosis, autophagy, and unfolded protein response; (d) M1 macrophage activation; (e) host mRNA transcription, cap-methylation, and stability; and (f) eIF2α dephosphorylation-mediated inhibition of viral protein translation. This study provides novel insights (summarized in Table 9) that warrant further investigation of VSV virulence factors and pathogenesis. In this context, further studies are encouraged to evaluate differences in the macrophage gene expression between the VSIV and VSNJV strains. Although there is some sequence conservation in some proteins between both strains, the glycoprotein conservation is as low as 50% [108]. A recent pathogenesis study conducted in pigs indicated that some VSIV strains might be as virulent as VSNJV strains [109]. The methodology described herein represents a good alternative to gain more insight into the difference in virulence between both serotypes. 

## 4. Materials and Methods 

### 4.1. Cell Culture of Macrophages and Viruses

Primary swine macrophage cell cultures were derived from pig peripheral blood as previously described [110]. Macrophages were seeded in 6-well plates (Primaria Falcon, Becton Dickinson, Franklin Lakes, NY, USA). The VSV strains used in this study included (a) NJ0612NME6, an epidemic VS New Jersey virus (VSNJV) strain that caused outbreaks in the US from 2012 to 2014 and was isolated from a naturally infected equine in New Mexico in 2012 and (b) NJ0806VCB, a VSNJV strain that circulated in 2006 in an endemic area of Mexico and was obtained from a naturally infected bovine in Veracruz [111]. These viruses have an overall nucleotide identity of 99.08%, and differences in their pathogenesis were reported in a previous study, which indicated that NJ0612NME6 has higher virulence than NJ0806VCB in pigs [12]. VSV infection experiments were conducted with three biological replicates using ex vivo cultured primary macrophages isolated from three different commercial domestic pigs. Macrophages were infected with an MOI of 10 TCID_50_ of each virus. Mock infection was also performed in the cultured macrophages from these same pigs as non-infected controls.

### 4.2. RNA Isolation 

Total RNA was extracted from primary swine macrophage cell cultures infected with the indicated viruses or mock-infected at 5 hours post-infection. Cells were harvested and lysed with a cell lysis buffer (Qiagen, Valencia, CA, USA), and RNA was isolated using a RNeasy mini kit (Qiagen, Valencia, CA, USA) according to the manufacturer’s instructions. The RNA quality was then determined using an Agilent 2100 bioanalyzer (Santa Clara, CA, USA) using an RNA nanochip, according to the procedures outlined by Agilent Technologies (Santa Clara, CA, USA). RNA was quantified using a Nanodrop 1000 (Thermo Scientific, Waltham, MA, USA).

### 4.3. DNA Microarray Analysis

A 44,000 (44K) porcine whole-genome expression microarray was designed based on pig expressed sequences (cDNA and EST) and porcine genome sequences homologous to non-porcine sequences, as reported by Zhu et al. [112]. All porcine EST and RNA sequences were downloaded from the NCBI database and assembled into unique sequences using the CAP3 software program [113]. The assembled sequences were aligned to pig genome sequences using the UCSC genome browser to select the 3′ end RNA sequences or the genome sequences aligned with other expressed sequences of other species if no porcine-expressed sequences were available. These selected sequences were used to design 60-mer oligonucleotide microarray probes with a low probability of cross-reacting with other genes and a bias to the 3′-end of RNA sequences using Array Designer 4.0 (Applied Biosystems, Foster City, CA, USA). Approximately 43K porcine probes were selected to synthesize a 44K Agilent microarray for this study. The annotation of the porcine sequences was based on the results of a BLAST search against human reference proteins and RNA sequences downloaded from NCBI databases and manual curation based on all expressed sequences aligned in the porcine genome sequences using the UCSC genome browser. One hundred and eighty-six duplicated probes designed from all ASFV open reading frames were also included in this custom microarray.

The custom-designed porcine microarrays were manufactured by Agilent Technologies and used for this study. Both ASFV-infected and mock-infected RNA samples were labeled with Cy3 and Cy5 individually using an Agilent low-input RNA labeling kit (Agilent Technologies). A Cy5-labeled ASFV-infected or mock-infected sample was co-hybridized with a Cy3-labeled mock-infected or ASFV-infected in one array, respectively, for each time point using a dye-swap design. The entire procedure of microarray analysis was conducted according to the protocols, reagents, and equipment provided or recommended by Agilent Technologies. Array slides were scanned using a GenePix 4000B scanner (Molecular Devices) with the GenePix Pro 6.0 software at 5 μM resolution.

### 4.4. Statistical and Bioinformatic Analyses of Microarray Data

Background signal correction and data normalization of the microarray signals and statistical analysis were performed using the LIMMA package [114]. Log_2_ fold changes in signal intensity were used in the statistical analysis to identify deferentially expressed genes. To account for multiple testing, the *p*-values were adjusted using the Benjamini and Hochberg method and expressed as a false discovery rate (FDR). The probe sequences were aligned to the porcine genome sequence displayed in the UCSC genome browser to validate the annotation by computational methods, such as BLAST. Gene expression differences with an FDR value of 0.05 or smaller and an expression difference of ≥50% were considered statistically significant and were considered differentially expressed genes (DEGs). Genes down- or upregulated in the infected macrophages compared to the non-infected macrophages were expressed as negative and positive values (fold), respectively. 

### 4.5. Pathway Analyses

The identified DEGs were mapped to human reference genes. Two lists of upregulated and downregulated genes associated with human Entrez gene IDs were analyzed with an NCBI online bioinformatics program (DAVID Bioinformatics Resources 6.8) to identify the biological pathways (GOTERM_BP_DIRECT, KEGG_PATHWAY, and REACTOME_PATHWAY) significantly over-represented by DEGs (*p* ≤ 0.05 with Benjamini correction). The DEGs with differential expressions of 2-fold or greater and 4-fold greater were used in these analyses to take the magnitudes of differentiation expression into consideration.

### 4.6. Biological Inference

The biological functions of DEGs in the identified over-represented pathways associated with the immune response were based on scientific publications obtained from PubMed. Biological inferences were based on (i) the immunological functions of the DEGs, (ii) gene expression levels based on microarray averaged signal intensity, and (iii) magnitudes (fold) of the differential expression, assuming higher mean signal intensity and larger differentially expressed genes play a bigger biological role in the gene groups. Genes with no significant differential expression but that are known to play important roles in the biological pathways associated with the significant DEGs were also used as supporting evidence. Genes down- or upregulated in the VSV-infected samples compared to the mock-infected samples were expressed as negative and positive values (fold), respectively. In this study, genes differentially expressed between infected and mock-infected macrophages were used to infer the molecular mechanisms of VSV pathogenesis and immune evasion.

## Figures and Tables

**Figure 1 pathogens-10-01134-f001:**
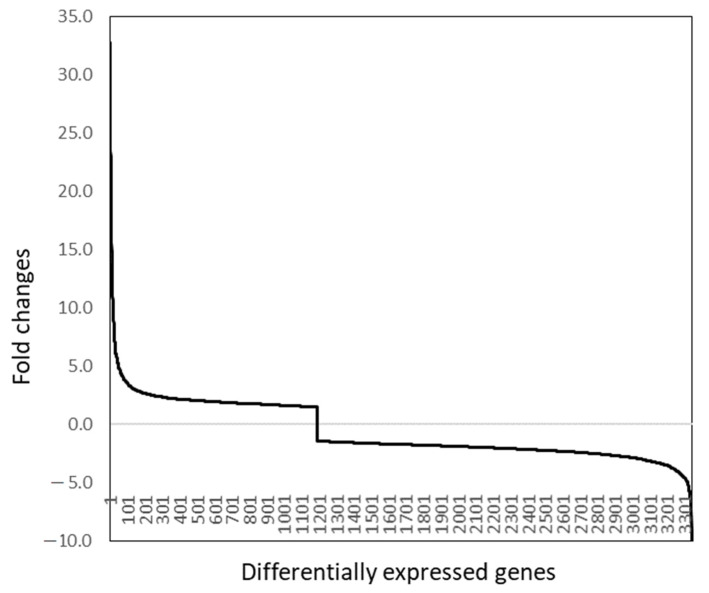
The distribution of fold changes of annotated genes with significantly differential expression equal to or greater than 2-fold between VSV- and mock-infected macrophages from most upregulated (32.8-fold) to most downregulated (–10.2-fold) genes in VSV-infected macrophages compared to mock-infection.

**Table 1 pathogens-10-01134-t001:** Gene ontology terms, Kyoto Encyclopedia of Genes, and Genomes (KEGG) and REACTOME biological pathways over-represented by genes differentially expressed by at least 2- and 4-fold between VSV-infected and mock-infected porcine macrophages using the NCBI DAVID program with Benjamini *p*-value correction.

	Pathway Analysis	Count	Benjamini
**DEGs with at Least 2-Fold Differential Expression**	GO_0007249: I-kappa B kinase/NF-kappa B signaling	23	8.2 × 10^5^
	GO_0016567: protein ubiquitination	62	0.026
	GO_0035666: TRIF-dependent toll-like receptor signaling pathway	12	0.036
	GO_0045944: positive regulation of transcription by RNA polymerase II	135	0.045
	KEGG_hsa04668: TNF signaling pathway	34	6.4 × 10^7^
	KEGG_hsa04620: Toll-like receptor signaling pathway	33	1.0 × 10^6^
	KEGG_hsa04064: NF-kappa B signaling pathway	28	5.4 × 10^6^
	KEGG_hsa04622: RIG-I-like receptor signaling pathway	21	5.7 × 10^4^
	KEGG_hsa04621: NOD-like receptor signaling pathway	16	0.011
	KEGG_hsa04144: Endocytosis	43	0.012
	KEGG_hsa05160: Hepatitis C	27	0.022
	KEGG_hsa05169: Epstein-Barr virus infection	25	0.026
	KEGG_hsa04210: Apoptosis	16	0.029
	KEGG_hsa05220: Chronic myeloid leukemia	17	0.043
	KEGG_hsa04140: Regulation of autophagy	9	0.050
	REACTOME_HSA-5357786: TNFR1-induced proapoptotic signaling	8	0.039
**DEGs with at Least 4-Fold Differential Expression**	GO_0006954: inflammatory response	19	1.2 × 10^5^
	GO_0006915: apoptotic process	20	5.4 × 10^4^
	GO_0045944: positive regulation of transcription by RNA polymerase II	25	0.003
	GO_0006955: immune response	15	0.009
	GO_0043065: positive regulation of apoptotic process	12	0.019
	GO_0071222: cellular response to lipopolysaccharide	8	0.020
	GO_0051897: positive regulation of protein kinase B signaling	7	0.021
	GO_0042981: regulation of apoptotic process	10	0.022
	GO_0070373: negative regulation of ERK1 and ERK2 cascade	6	0.026
	GO_2001244: positive regulation of intrinsic apoptotic signaling pathway	5	0.029
	GO_0030593: neutrophil chemotaxis	6	0.036
	GO_0051384: response to glucocorticoid	6	0.037
	GO_0070098: chemokine-mediated signaling pathway	6	0.047
	KEGG_hsa04060: Cytokine–cytokine receptor interaction	15	3.8 × 10^5^
	KEGG_hsa04668: TNF signaling pathway	11	4.7 × 10^5^
	KEGG_hsa04064: NF-kappa B signaling pathway	7	0.016
	KEGG_hsa04621: NOD-like receptor signaling pathway	6	0.020
	KEGG_hsa04620: Toll-like receptor signaling pathway	7	0.027
	Reactome-HSA-380108: Chemokine receptors bind chemokines	7	0.003

**Table 2 pathogens-10-01134-t002:** Expression levels (EXP), false discovery rates (FDR), and fold differences (epidemic vs. mock infection: EP/M, epidemic vs. endemic infection: EP/EN) of interferon-signaling genes differentially expressed between infected- and/or mock-infected macrophages.

Group	Gene	EXP	EP/M	FDR	EP/EN	FDR
**IFN and Signaling**	IFNA17	77	−1.9	0.01	−1.1	0.96
	IFNB	179	6.2	0.04	−2.1	0.79
	IFNAR1	3774	−2.0	0.01	1.0	0.97
	IFNAR2	2223	−2.9	0.03	−1.2	0.90
	IFNGR1	1000	−2.7	0.02	−1.2	0.92
**Ten Typical Interferon Stimulated Genes**	IFI44L	955	1.1	0.88	1.2	0.82
	IFIH1	107	−1.1	0.92	−1.0	1.00
	IFIT1	231	1.7	0.52	−1.4	0.89
	IFIT2	420	−2.4	0.23	1.0	0.99
	IFIT3	3298	−1.6	0.27	1.2	0.89
	IFIT5	364	−2.0	0.10	1.4	0.80
	ISG20	580	1.2	0.91	1.3	0.96
	MX1	3882	−1.2	0.90	−1.0	0.99
	MX2	1330	−1.5	0.56	1.2	0.94
	OAS1	155	−1.1	0.95	1.1	0.98
**IFN Expression Inhibitors**	AHR	193	6.1	0.01	−1.0	0.98
	ATF3	1432	8.6	0.05	−1.6	0.91
	DUSP1	2208	13.8	0.01	−1.6	0.88
	EGR1	891	13.4	0.01	−2.1	0.79
	FOS	1993	22.9	0.01	−1.7	0.90
	HES1	218	10.6	0.02	−1.2	0.97
	PRDM1	796	32.8	0.01	−1.9	0.86

**Table 3 pathogens-10-01134-t003:** Expression levels (EXP), false discovery rates (FDR), and fold differences (epidemic vs. mock infection: EP/M, epidemic vs. endemic infection: EP/EN) of interferon expression regulating genes differentially expressed between infected- and/or mock-infected macrophages.

Pathway	Gene	EXP	EP/M	FDR	EP/EN	FDR
**MAPK**	ATF2	300	−2.4	0.01	1.3	0.75
	MAP2K5	1804	−1.7	0.02	1.0	0.99
	MAPK14/p38	539	−3.2	0.03	1.0	1.00
	MAPK4	176	−2.8	0.02	1.4	0.79
	MAP3K18	2279	−4.3	0.00	1.1	0.98
**NFκB**	CARD6	404	−2.6	0.01	1.1	0.95
	IKBKB	907	−2.2	0.01	−1.0	0.99
	IRAK1	5234	−3.5	0.00	−1.0	0.98
	NLK/NEMO	170	−3.3	0.01	1.2	0.88
	TAB1	1817	−2.2	0.03	1.1	0.98
	TAB2	416	−2.7	0.00	1.1	0.91
	TAK1	440	−2.1	0.01	1.2	0.80
	NFKBIA	10873	7.9	0.01	−1.2	0.95
	NFKBID	1182	4.2	0.03	−1.6	0.81
	TNFAIP3/A20	799	8.3	0.00	−1.4	0.88
**RIG-I**	IRF5	3796	−2.1	0.01	−1.1	0.95
	MAVS	452	−2.1	0.03	−1.1	0.95
	TBK1	459	−2.0	0.05	−1.1	0.97
**Toll-like Receptor**	BTK	4636	−2.0	0.01	1.2	0.82
	TLR1	361	−2.8	0.02	1.2	0.93
	TLR2	11817	−3.8	0.00	−1.1	0.94
	TLR4	543	−3.8	0.01	1.0	0.99
	TICAM2/TRIF	129	−2.8	0.02	1.3	0.84
	TLR6	143	−3.3	0.01	1.2	0.91
	TLR7	293	1.7	0.03	1.0	0.99

**Table 4 pathogens-10-01134-t004:** Expression levels (EXP), false discovery rates (FDR), and fold differences (epidemic vs. mock infection: EP/M, epidemic vs. endemic infection: EP/EN) of cytokine, chemokine, and the receptor genes differentially expressed between infected- and/or mock-infected macrophages.

Group	Gene	EXP	EP/M	FDR	EP/EN	FDR
**Cytokines**	CSF3	750	4.0	0.05	−1.3	0.93
	IL1A	698	13.9	0.02	−1.9	0.87
	IL1B	1793	7.7	0.13	−1.7	0.92
	IL10	1109	11.5	0.00	−1.9	0.76
	IL27	318	3.8	0.01	−1.9	0.67
	TNF	2641	23.4	0.01	−1.7	0.87
	TNFSF9/CD137L	369	5.2	0.02	−2.4	0.66
	TNFSF11	729	−4.5	0.02	1.2	0.94
	AREG	258	17.5	0.01	−1.6	0.91
	HBEGF	673	4.7	0.04	−1.3	0.94
	LIF	217	12.9	0.01	−1.9	0.79
	VEGFA	384	5.7	0.03	−1.6	0.86
**Cytokine Receptors**	IL1R2	219	2.9	0.04	−1.3	0.88
	IL17RA	4899	−1.7	0.02	1.0	0.98
	IL20RB	92	2.7	0.03	−1.4	0.80
	LTBR	8211	−3.5	0.01	1.1	0.95
	TNFRSF1A	3943	−2.3	0.01	−1.0	1.00

**Table 5 pathogens-10-01134-t005:** Expression levels (EXP), false discovery rates (FDR), and fold differences (epidemic vs. mock infection: EP/N, epidemic vs. endemic infection: EP/EN) of chemokine and the receptor genes differentially expressed between infected- and/or mock-infected macrophages.

Group	Gene	EXP	EP/M	FDR	EP/EN	FDR
**CCLs**	CCL3	2336	7.4	0.01	−1.8	0.80
	CCL4	3028	29.1	0.00	−2.1	0.79
	CCL5	6487	3.1	0.06	−1.4	0.88
	CCL5_v	385	23.0	0.01	−1.6	0.90
	CCL20	412	17.7	0.01	−1.4	0.94
**ELR+ CXCLs**	CXCL1	1791	6.4	0.03	−1.1	0.98
	CXCL2	3008	21.2	0.00	−1.9	0.81
	CXCL3	2388	6.8	0.02	−1.5	0.90
**CCL/CXCL Receptors**	CCR5	246	−2.8	0.02	−1.0	1.00
	CCR7	206	4.5	0.00	−1.3	0.84
	CCRL2	1839	−1.8	0.02	1.1	0.94
	CX3CR1	129	−2.2	0.02	1.1	0.92
	CXCR4	1060	4.3	0.02	−1.1	0.98

**Table 6 pathogens-10-01134-t006:** Expression levels (EXP), false discovery rates (FDR), and fold differences (epidemic vs. mock infection: EP/M, epidemic vs. endemic infection: EP/EN) of apoptosis-, autophagy-, and unfold protein response (UPR)-related genes differentially expressed between infected- and/or mock-infected macrophages.

Group	Gene	EXP	EP/MM	FDR	EP/EN	FDR
**Apoptosis and Death Receptor Signaling**	BCL2L13	1167	−2.0	0.01	1.2	0.77
	CASP8	337	−2.1	0.02	1.1	0.90
	DAPK1	246	−2.2	0.01	1.1	0.96
	DIDO1	162	−2.1	0.02	1.3	0.75
	BIRC3/cIAP2	656	3.5	0.01	−1.2	0.90
	REL	100	4.8	0.02	−2.4	0.66
	SGK1	2392	4.1	0.01	−1.4	0.86
	BRE	184	2.9	0.05	−2.0	0.68
	FADD	145	−2.2	0.04	−1.0	1.00
	IER3	988	5.6	0.02	−1.5	0.87
	RIPK1	1864	−2.1	0.01	−1.1	0.91
	TRADD	3504	−2.1	0.03	−1.0	0.98
**Autophagy**	ATG3	2041	−1.8	0.03	1.1	0.92
	ATG4B	1154	−2.8	0.01	1.0	0.99
	ATG5	524	−2.1	0.05	1.2	0.88
	ATG9A	977	−2.2	0.00	1.1	0.96
	ATG16L1	351	−2.3	0.01	−1.3	0.72
	ATG16L2	1146	−2.1	0.00	−1.1	0.95
	ATG101	1026	−2.7	0.01	−1.1	0.95
	FLCN	771	−4.8	0.01	−1.1	0.97
	RAB33B	241	−3.0	0.00	1.4	0.70
	RB1CC1	147	−1.9	0.03	1.1	0.97
	ULK1	1083	−1.8	0.03	−1.1	0.96
	BCL2L11/BIM	152	4.3	0.00	−1.6	0.71
	GADD45B	649	22.6	0.00	−1.6	0.86
**UPR**	ERN1	146	−2.6	0.02	1.0	0.99
	PPP1R15A	850	7.0	0.01	−1.5	0.85
	XBP1	4634	−6.1	0.01	1.1	0.98

**Table 7 pathogens-10-01134-t007:** Expression levels (EXP), false discovery rates (FDR), and fold differences (epidemic vs. mock infection: EP/M, epidemic vs. endemic infection: EP/EN) of transcription- and translation-related genes differentially expressed between infected- and/or mock-infected macrophages.

Group	Gene	EXP	EP/M	FDR	EP/EN	FDR
mRNA transcription	CDK7	359	−3.1	0.01	1.2	0.91
	GTF2A1	534	−2.5	0.00	1.3	0.72
	GTF2B	1429	−2.2	0.05	1.2	0.92
	GTF2E1	176	−2.2	0.02	1.1	0.97
	MNAT1	217	−1.7	0.03	1.2	0.84
	RPAP3	161	−1.6	0.05	1.0	0.99
	TAF1C	481	−3.1	0.03	−1.2	0.94
	TAF7	716	−2.7	0.01	1.2	0.89
	TAF11	882	−3.2	0.03	1.1	0.97
	TBP	1191	−1.8	0.01	1.2	0.92
	TCEANC2	130	−2.6	0.01	1.3	0.75
	TCF20	348	−2.2	0.03	−1.0	0.98
	TFCP2	248	−1.9	0.01	1.1	0.91
mRNA processing	CMTR2	180	−3.5	0.01	1.3	0.83
	DICER1	1024	−2.4	0.02	−1.1	0.98
	MIR132	78	3.3	0.03	−2.7	0.54
	TIAL1	93	−3.7	0.04	−1.3	0.88
	ZFP36/TTP	5049	4.5	0.03	−2.0	0.74

**Table 8 pathogens-10-01134-t008:** Expression levels (EXP), false discovery rates (FDR), and fold differences (epidemic vs. mock infection: EP/M, epidemic vs. endemic infection: EP/EN) of genes that are associated with macrophage immunity and were differentially expressed between infected- and/or mock-infected macrophages.

Group	Gene	EXP	EP/M	FDR	EP/EN	FDR
Inflammation mediator	ADM	331	5.9	0.01	−1.3	0.90
	DUSP2	2264	3.8	0.01	−1.0	0.99
	MALAT1	125	6.0	0.01	−1.7	0.75
	PTGS2	300	15.6	0.02	−1.3	0.95
Macrophage activation	C5AR1	10355	−3.0	0.01	1.1	0.94
	IL4I1	297	3.6	0.01	−1.2	0.89
	P2RY1	505	−7.3	0.00	1.1	0.96
	P2RY6	290	−2.6	0.05	1.1	0.97
Immune stimulators	MAP3K8	779	10.0	0.00	−1.5	0.85
	MEFV	1158	6.7	0.03	−1.7	0.85
	CEBPD	2210	−9.3	0.02	−1.1	0.99
	DUSP6	4285	−6.3	0.01	1.0	1.00
	MAFB	3002	−10.0	0.00	−1.3	0.91
	MAPK8IP3/JIP3	1932	−5.8	0.01	1.0	1.00
	MTRES1	1804	−9.7	0.00	1.2	0.93

**Table 9 pathogens-10-01134-t009:** Differentially expressed genes used to infer candidate mechanisms of VSV systemic and tissue pathogenesis and immune evasion in infected and non-infected cells.

Pathogenesis	Mechanism Inferred from Differentially Expressed Genes
**Systemic**	Fever: ↑ CCLs [3, 4, 5], CXCLs [1, 2, 3], IL1A, PGTS2, TNF
	Anorexia: ↑ LIF, PGTS2
	Systemic infection restriction: ↑ IFNB, VSV sensitive to IFN inhibition
**Local Tissue**	Immune cell infiltration: ↑ CCLs [3, 4, 5, 20], CXCLs [1, 2, 3], PTGS2
	Inflammation: ↑ ADM, PGTS2
	Vasodilatation: ↑ PGTS2
	Necroptosis/apoptosis: ↑ TNF, ↓ TIAL1
**Infected/Non-infected Cells**	↓ General immune response: ↑ IL10
	↓ Interferon response: ↑ AREG, HBEGF, VEGF, IL1A
	↓ MΦ, neutrophils, Th1, CTL, NK cell activities: ↑ PGE/PGTS2
	↓ Th17 response: ↑ LIF; ↓ IL17RA
**Infected Cells**	↓ IFNB production: ↑ AHR, ATF3, DUSP1, FOS, HES1, PRDM1; ↓ ATF2, XBP1
	↓ MAPK signaling: ↓ MAPK4, MAPK14/p38, MAP3K18
	↓ RIG-I signaling: ↓ IRF5, MAVS, TBK1
	↓ NFκB signaling: ↑ NFKBIA, NFKBID, A20; ↓ TNFSF11, 7 DEGs
	↓ TLR4 signaling: ↓ BTK, TICAM2, TLR4
	↓ Interferon signaling: ↓ IFNAR1, IFNAR2, IFNGR1, ↑ ATF3
	↓ Apoptosis and/or autophagy: ↑ GADD45B, ↓ 18 DEGs (Table 6)
	↓ TNF signaling: ↓ FADD, RIPK1, TNFRSF1A, TRADD; ↑ BRE, IER3
	↓ Unfolded protein response: ↓ ERN1, XBP1, ↑ GADD34
	↓ AT-rich (cytokine) mRNA stability: ↑ ZFP36
	↓ Host mRNA cap-methylation/translation: ↓ CMTR2
	↓ Host transcription: ↓ 13 DEGs involved in RNA transcription (Table 7)
	↓ MΦ M1 activation: ↓C5AR1, CCR5, IL17RA, Y2RY1, Y2RY6; ↑IL4I1, TNFSF9
	↑ VSV replication (↓ ISG expression): ↑MIR132
	↑ VSV replication (unknown mechanisms): ↓ DICER1, ↑ TLR7
	↑ VSV protein synthesis via eIF2α dephosphorylation: ↑ GADD34

## Data Availability

The microarray raw data are in the process of being submitted to the NCBI Gene Expression Omnibus (GEO) database. The data sets will be available to the public if this manuscript is accepted for publication.

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
