# Peer review of "Molecular Pathogenesis and Immune Evasion of Vesicular Stomatitis New Jersey Virus Inferred from Genes Expression Changes in Infected Porcine Macrophages"

_pathogens, 2021, doi:10.3390/pathogens10091134_

Round 1
Reviewer 1 Report
Macrophages represent an important component of the immune response to VSV infection in livestock. But the changes induced in macrophages by VSV are not yet well characterized. In this manuscript, the transcriptomic changes induced by VSV infection of primary cultured porcine macrophages are examined through a microarray comparison of the expression patterns in uninfected cells vs. that in cells infected with either an endemic or epidemic strain of VSV. The study identifies a plethora of both upregulated and downregulated genes that provide much-needed insight into both the role of macrophages in the response to the virus and how the latter manipulates this response for its own advantage. The data provide strong confirmation that VSV suppresses multiple components of the immune response including IFN beta expression; type I and II interferon, IL-1 and TLR and RIG-I signaling pathways; processes involved in apoptosis and autophagy; and host mRNA transcription, cap-methylation and stability and protein translation, the latter through dephosphorylation of eIF2a. Many of the genes are significantly upregulated in the VSV-infected macrophages by more than 10-fold, including most of the IFN-expression inhibitors, as well as the cytokines IL1A, !L10 and TNF and the receptor genes CCL4, CCL5_v, CCL20 and CXCL2. Overall, the data confirm that VSV suppresses the interferon response in a multi-faceted way.
This is considered a very strong manuscript that contributes a clear and comprehensive understanding of the breadth and scope of the strategies used by VSV to control the macrophage’s ability to respond to the virus. The authors present an enormous amount of data in a well-organized and concise manner. In addition, Table 9 provides a perfect summary of the most significant findings of the manuscript and one can easily envision the findings summarized in this table serving as a springboard for numerous studies to come. With the exception of some very minor points, as detailed below, the manuscript is considered strong enough for publication once these points are addressed in a revision.
Minor points
Line 175: Changes in the expression of 6 activator genes are discussed here. However, there are 7 such genes listed.
Line 189: Here, VSV is said to significantly induce the expression 5 immune cytokines, but then 6 are listed.
Lines 272-4: There is something wrong with these sentences.
Lines 320-1: inhibit of?
Line 384: The word “chemokines” stands alone here.
Author Response
Dear reviewer one, thank you for the positive comments to our study. We appreciate your suggestions to improve the quality of this manuscript. Below, you will find the responses to your observations.
Line 175: Changes in the expression of 6 activator genes are discussed here. However, there are 7 such genes listed.
Response: this was corrected.
“Likewise, the expression of seven activator genes (CARD6, IKBKB, IRAK1, NLK, TAB1, TAB2, and TAK1) in the NFκB pathway was significantly downregulated in VSV-infected cells, whereas the expression of three inhibitors of NFκB [NFKBIA, NFKBID, and TNFIP3/A20 (38) was significantly upregulated (Table 3).”
Line 189: Here, VSV is said to significantly induce the expression 5 immune cytokines, but then 6 are listed.
Response: this was corrected.
“VSV infection significantly induced the expression of six immune cytokines (CSF3, IL1A, IL10, IL27, TNF and TNFSF9) and suppressed TNFSF11 expression (Table 4).”
Lines 272-4: There is something wrong with these sentences.
Response: The sentence was rephrased see lines 272-5
On the other hand, four proinflammatory genes (CEBPD, DUSP6, MAFB, and MAPK8IP3/JIP3) associated with the regulation of NF-kb pathway (66, 67, 68, 69), and one gene (MTRES1) implicated in the stress response in mitochondria (70), were downregulated during the VSV infection.
Lines 320-1: inhibit of?
Response: This was corrected.
“TNF leads to IL-10 production by monocytes and together with IL-10, inhibit CD4 T-cell expansion and function (84).”
Line 384: The word “chemokines” stands alone here.
Response: This word was eliminated.
Reviewer 2 Report
The manuscript by Velazquez-Salinas et al., reports the comprehensive gene expression profile in porcine macrophages. The manuscript is well written, study is well designed and executed with results presented clearly.
Specific comment:
1) In the title of the manuscript, please include the new jersey strain saying VSV NJ strain. Since the strain difference has been shown to have different pathogenicity.
2) Please include the genetic sequence differences between the strains used in this study and discuss the difference in responses seen with respect to the genetic variability.
3) Please cite and discuss this manuscript " Comparative evaluation of pathogenicity of three isolates of vesicular stomatitis virus (Indiana serotype) in pigs" Morozov et al., Journal of General Virology 2019;100:1478–1490 DOI 10.1099/jgv.0.001329. How different are these strains compared to the NJ strain?
Author Response
Dear reviewer two, thank you for the positive comments to our study. We appreciate your suggestions to improve the quality of this manuscript. Below, you will find the responses to your observations.
Specific comment:
1) In the title of the manuscript, please include the new jersey strain saying VSV NJ strain. Since the strain difference has been shown to have different pathogenicity.
Response: We agree with your observation. The title of this manuscript was modified: “Molecular Pathogenesis and Immune Evasion of Vesicular Stomatitis New Jersey Virus Inferred from Genes Expression Changes in Infected Porcine Macrophages
2) Please include the genetic sequence differences between the strains used in this study and discuss the difference in responses seen with respect to the genetic variability.
Response: Information about the differences between both strains was included both in the methods (lines 443-4) and discussion sections (Lines 417-8). This is discussed in lines 417-424. New reference was added to support our statement.
Martinez, I., Rodriguez, L.L., Jimenez, C., Pauszek, S.J., Wertz, G.W., 2003. Vesicular stomatitis virus glycoprotein is a determinant of pathogenesis in swine, a natural host. J Virol 77, 8039-8047.
3) Please cite and discuss this manuscript " Comparative evaluation of pathogenicity of three isolates of vesicular stomatitis virus (Indiana serotype) in pigs" Morozov et al., Journal of General Virology 2019;100:1478–1490 DOI 10.1099/jgv.0.001329. How different are these strains compared to the NJ strain?
Response: The suggested reference was added (reference 109) and discussed in lines 432-438. Two additional references were added to support differences in pathogenesis between NJ and Ind strains.
Martinez, I., Rodriguez, L.L., Jimenez, C., Pauszek, S.J., Wertz, G.W., 2003. Vesicular stomatitis virus glycoprotein is a determinant of pathogenesis in swine, a natural host. J Virol 77, 8039-8047.
Martinez I, Wertz GW. Biological differences between vesicular stomatitis virus Indiana and New Jersey serotype glycoproteins: identification of amino acid residues modulating pH-dependent infectivity. J Virol. 2005;79(6):3578-85.